# Task-Agnostic Morphology Evolution

**Donald J. Hejna III**
UC Berkeley
jhejna@berkeley.edu

**Pieter Abbeel**
UC Berkeley
pabbeel@berkeley.edu

**Lerrel Pinto**
New York University
lerrel@cs.nyu.edu

## Abstract

Deep reinforcement learning primarily focuses on learning behavior, usually over-looking the fact that an agent's function is largely determined by form. So, how should one go about finding a morphology fit for solving tasks in a given environment? Current approaches that co-adapt morphology and behavior use a specific task's reward as a signal for morphology optimization. However, this often requires expensive policy optimization and results in task-dependent morphologies that are not built to generalize. In this work, we propose a new approach, Task-Agnostic Morphology Evolution (TAME), to alleviate both of these issues. Without any task or reward specification, TAME evolves morphologies by only applying randomly sampled action primitives on a population of agents. This is accomplished using an information-theoretic objective that efficiently ranks agents by their ability to reach diverse states in the environment and the causality of their actions. Finally, we empirically demonstrate that across 2D, 3D, and manipulation environments TAME can evolve morphologies that match the multi-task performance of those learned with task supervised algorithms. Our code and videos can be found at https://sites.google.com/view/task-agnostic-evolution.

## 1 Introduction

Recently, deep reinforcement learning has shown impressive success in continuous control problems across a wide range of environments (Schulman et al., 2017; Barth-Maron et al., 2018; Haarnoja et al., 2018). The performance of these algorithms is usually measured via the reward achieved by a pre-specified morphology on a pre-specified task. Arguably, such a setting where both the morphology and the task are fixed limits the expressiveness of behavior learning. Biological agents, on the other hand, both adapt their morphology (through evolution) and are simultaneously able to solve a multitude of tasks. This is because an agent's performance is intertwined with its morphology as morphology fundamentally endows an agent with the ability to act. But how should one design morphologies that are performative across tasks?

Recent works have approached morphology design using alternating optimization schemes (Hazard et al., 2018; Wang et al., 2019; Luck et al., 2020). Here, one step evaluates the performance of morphologies through behavior optimization while the second step improves the morphology design typically through gradient-free optimization. It thus follows that the final morphology's quality will depend directly on the quality of learned behavior as inadequate policy learning will result in a noisy signal to the morphology learner. This begs the question: is behavior learning a necessary crutch upon which morphology optimization should stand?

Unfortunately, behavior learning across a multitude of tasks is both difficult and expensive and hence a precise evaluation of each new candidate morphology requires explicit policy training. As a result, current research on morphology optimization primarily focuses on improving morphology for just one task (Wang et al., 2019; Ha, 2019). By exploiting task-specific signals, learned morphologies demonstrate impressive performance but provide no guarantees of success outside of the portion of the environment covered by the given task. This is at odds with biological morphologies that are usually able to complete many tasks within their environment. Fundamentally, we want agents that are generalists, not specialists and as such, we seek to shift the paradigm of morphology optimization to multi-task environments. One obvious solution to this issue would be to just learn multiple behaviors in the behavior-learning step. However, such an approach has two challenges. First, multi-task RL is both algorithmically and computationally inhibiting, and hence in itself is an active area of research

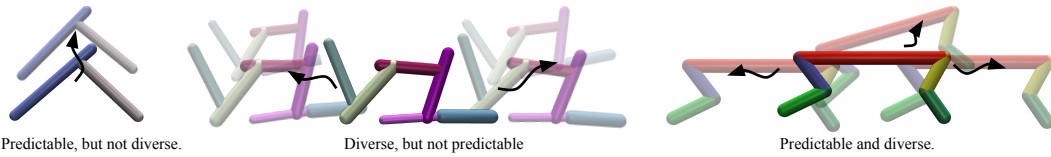

Predictable, but not diverse.      Diverse, but not predictable      Predictable and diverse.

*Figure 1.* Visual motivation of our information theoretic objective. We seek to evolve morphologies that can easily explore a large number of states and remain predicable while doing so.

(Fu et al., 2016; Yu et al., 2020). Second, it is unrealistic to assume that we can enumerate all the tasks we would want an agent to perform before its inception.

In this work, we propose a framework for morphology design without the requirements of behavior learning or task specification. Instead, inspired by contemporary work in unsupervised skill discovery (Eysenbach et al., 2018; Sharma et al., 2019) and empowerment (Mohamed & Rezende, 2015), we derive a task-agnostic objective to evaluate the quality of a morphology. The key idea behind this evaluator is that a performant morphology is likely one that exhibits strong exploration and control by easily reaching a large number of states in a predictable manner. We formalize this intuition with an information-theoretic objective and use it as a fitness function in an evolutionary optimization loop. Candidate morphologies are mutated and then randomly sample and execute action primitives in their environment. The resulting data is used to estimate the agents' fitness per the information-theoretic objective.

Our contributions are summarized as follows: First, we derive an easily computable information-theoretic objective to rank morphologies by their ability to explore and control their environment. Second, using this metric in conjunction with Graph Neural Networks, we develop Task-Agnostic Morphology Evolution (TAME), an unsupervised algorithm for discovering morphologies of an arbitrary number of limbs using only randomly sampled action primitives. Third, we empirically demonstrate that across 2D, 3D, and manipulation environments TAME can evolve morphologies that match the multi-task performance of those learned with task supervised algorithms.

## 2 RELATED WORK

Our approach to morphology optimization builds on a broad set of prior work. For conciseness, we summarize the most relevant ones.

**Morphology Optimization.** Optimizing hardware has been a long studied problem, yet most approaches share two common attributes: first, they all focus on a single task, and second, they all explicitly learn behavior for that task. Sims (1994) pioneered the field of morphology optimization by simultaneously evolving morphologies of 3D-blocks and their policy networks. Cheney et al. (2013) and Cheney et al. (2018) reduce the search space by constraining form and function to oscillating 3D voxels. More recently, Nygaard et al. (2020) evolve the legs of a real-world robot. Unlike TAME, these approaches depend on task reward as a fitness function to maintain and update a population of agents. Quality diversity based objectives (Lehman & Stanley, 2011; Nordmoen et al., 2020) augment regular task-fitness with unsupervised objectives to discover a diverse population of agents. These approaches are complementary to ours as quality diversity metrics could be incorporated into the TAME algorithm for similar effects. RL has also been applied to optimize the parameters of an agent's pre-defined structure. Ha (2019) use a population-based policy-gradient method, Schaff et al. (2019) utilize a distribution over hardware parameters, Luck et al. (2020) learn a morphology conditioned value function, and Chen et al. (2020) treat hardware as policy parameters by simulating the agent with computational graphs. While these RL-based approaches explicitly learn task behavior to inform morphology optimization, we do not learn any policies due to their computation expense. Moreover, all these methods are gradient-based, restricting them to fixed topology optimization where morphologies cannot have a varying number of joints.

**Graph Neural Networks.** Graph Neural Networks have shown to be effective representations for policy learning across arbitrary agent topologies (Wang et al., 2018; Huang et al., 2020). These representations have also been used for agent design. Pathak et al. (2019) treats agent construction as an RL problem by having modular robots learn to combine. Most related to our work, Neural Graph

Evolution (NGE) Wang et al. (2019) evolves agents over arbitrary graph structures by transferring behavior policies from parent to child. Unlike other RL approaches, the use of graph networks allows NGE to mutate arbitrary structures. While these works again are task supervised and only learn morphology for forward locomotion, they inform our use of GNNs to estimate our learning objective.

**Information Theory and RL.** Our information theoretic objective is inspired by several recent works at the intersection of unsupervised RL and information theory. Eysenbach et al. (2018) and Sharma et al. (2019) both use information theoretic objectives in order to discover state-covering skills. We apply similar logic to the problem of discovering state-covering morphologies. Gregor et al. (2016), Mohamed & Rezende (2015) and Zhao et al. (2020) estimate a quantity called "empowerment" (Klyubin et al., 2005) for intrinsic motivation. While empowerment maximizes the mutual information between final state and a sequence of actions by changing actions, we optimize the mutual information over morphologies. Additionally, these metrics for intrinsic motivation require policy learning. More broadly, Oord et al. (2018) use mutual information to learn representations in an unsupervised manner.

## 3 METHOD

In this section we introduce our algorithm for Task Agnostic Morphology Evolution (TAME). TAME works by ranking morphologies of arbitrary topology by an information theoretic quantity that serves as proxy for how well an agent can explore and control its environment. Practically, this is accomplished using a GNN to predict the action primitive a morphology executed to reach a specific state. By progressively mutating agents of high fitness according to this metric, TAME discovers morphologies functional over a large portion of the environment without task specification.

### 3.1 A TASK-AGNOSTIC OBJECTIVE FOR FITNESS

In order to accomplish any task in its environment, a morphology should be able to reliably reach any state through a unique sequence of actions. For example, a morphology that can reach many states but does so stochastically is uncontrollable, while a morphology that can visit a diverse set of states as a consequence of its actions is empowered. We capture this intuition using information theory. Let $S, S_T, A,$ and $M$ be random variables representing starting state, terminal state, action primitive, and morphology respectively. Our overall objective is to find morphologies that exhibit high mutual information between terminal states and action primitives, $I(S_T; A|S) = H(S_T|S) - H(S_T|A, S)$. Concretely, starting at state $S$, a good morphology ought to be able to visit a large number of terminal states or equivalently have high entropy $H(S_T|S)$. Second, the attained terminal state $S_T$ ought to be easily predicted given the action primitive taken $A$, or $H(S_T|A, S)$ should be low. As we seek to find morphologies that innately maximize this quantity, our objective becomes $\arg\max_m I(S_T; A|S, M = m)$. By assuming that all morphologies begin in the same state, we remove the objective's dependence on starting state $S$ and derive a variational lower bound as in (Barber & Agakov, 2003).

$$\arg\max_m I(S_T; A|M = m)$$
$$= \arg\max_m H(A|M = m) - H(A|S_T, M = m)$$
$$\geq \arg\max_m H(A|M = m) + \mathbb{E}_{a \sim p(A|m), s_T \sim p(S_T|a,m)}[\log q_\phi(a|s_T, m)]$$

Note that in the first line we take the dual definition of mutual information, that a morphology $M$ should be able to take as many actions as possible and each of those actions should be predictable from the final state $S_T$. The action distribution depends on the size of the action space of the morphology, thus $a \sim p(A|m)$ and the dynamics also depends on the morphology, thus $s_T \sim p(S_T|a, m)$. We then attain a lower bound on our objective by using a classifier $q_\phi(a|s_T, m)$ to predict the action primitive taken given a morphology and final state. However, as written this objective still presents a problem: different morphologies have different action spaces, and thus would require different action primitives. We resolve this issue by assuming that each morphology $m$ is composed of $k^{(m)}$ joints and that every joint has the same possible set of action primitives. Thus an overall action primitive can be denoted as $A = \{A_1, ..., A_{k^{(m)}}\}$ and we denote $|A_j|$ as the number of possible primitives for joint $j$. If we take each joint's action primitive to be independently sampled from a uniform

distribution, we can reduce the $H(A|M)$ term. We get:

$$\arg\max_m H(A|M=m) + \mathbb{E}_{a \sim p(A|m), s_T \sim p(S_T|a,m)}[\log q_\phi(a|s_T, m)]$$

$$= \arg\max_m k^{(m)}(\log|A_j| + (1/k^{(m)})\mathbb{E}_{a \sim p(A|m), s_T \sim p(S_T|a,m)}[\log q_\phi(a|s_T, m])$$

$$\geq \arg\max_m (k^{(m)})^\lambda (\log|A_j| + (1/k^{(m)})\mathbb{E}_{a \sim p(A|m), s_T \sim p(S_T|a,m)}[\log q_\phi(a|s_T, m)]) \quad (1)$$

where $0 \leq \lambda \leq 1$. A full derivation can be found in Appendix A. The resulting objective has two terms. The latter can be interpreted as the average mutual information between the actions of each joint $A_j$ and the final state $S_T$. The former can be interpreted as an adjustment to the overall morphology's information capacity based on the size of its action space or number of joints. Left untouched the objective would grow linearly in number of joints, but logarithmically in the accuracy of $q_\phi$. As later detailed in section 3.3, we only want the best morphologies to have high classification accuracy. As such, we introduce a regularizer $\lambda$ to attenuate the effect of adding more limbs and emphasize predictability. Similar to empowerment, our objective uses the mutual information between states and actions. However rather than conditioning on starting state and maximizing over a sequence of actions, we maximize with respect to morphology. By assuming the distribution of joint action primitives is uniform, we assume that morphologies ought to use all abilities endowed by their structure. In the next section, we detail how we practically estimate this objective over a set of morphologies.

## 3.2 Practically Estimating the Objective

In order to estimate the objective in equation 1, we need (a) a representation of morphologies that is able to capture arbitrary topologies, (b) a set of action primitives common across every joint, and (c) a classifier $q_\phi(a|s_t, m)$ that is able to predict said action primitives. We simultaneously address these challenges using graph representations as follows:

**Morphology Representation.** To represent arbitrary morphologies, we follow Sanchez-Gonzalez et al. (2018) and Wang et al. (2019) and represent each morphology $m$ by a tree $T = (E, V)$, where the vertices $V$ represent limbs and the edges $E$ represent the joints between them. By constructing vertex embeddings $v_i$ that contain the radius, extent, and position of each limb and edge embeddings $e_k$ for each limb-connection containing the joint type and range, we define a one-to-one mapping between trees $T$ and morphologies $M$.

**Action Distribution.** Successes in Deep RL (Mnih et al., 2013) have shown that random actions offer sufficient exploration to find solutions to complex tasks. Consequently, we posit that randomly sampled action sequences applied to each of a morphology's joints should be enough to estimate its fitness. However, purely random actions would provide no usable signal to $q_\phi$. Thus, we define each joint's action space to be a set of predefined action primitives that can each be executed for an extended number of timesteps. In our experiments this takes the form of sinusoidal inputs with different frequencies and phases or constant torques of differing magnitudes.

**GNN Classifier.** In order to be able to predict the action primitive taken by each joint of varying sized morphologies, we represent $q_\phi$ as a graph neural network classifier trained with cross entropy loss. As we apply actions to joints rather than limbs, we preprocess each morphology tree $T$ by transforming it to its line graph $G_l$, where each vertex in $G_l$ is given by an edge in $E$. For edge $(i, j) \in E$, we set the embedding of the line graph vertex to the concatenation $v^l_{(i,j)} = [v_i, v_j, e_{(i,j)}]$, where $v_i, v_j$ are the original node embeddings and $e_{(i,j)}$ is the original edge embedding. As we only care about how an agent influences its environment and not its internal state, we also concatenate environment state $s_T$ to each of the node embeddings in $G_l$. The label for each node in $G_l$ is the action taken by the corresponding joint. We implement $q_\phi$ using Graph Convolutions from Morris et al. (2019). More details can be found in Appendix G. We then estimate $\mathbb{E}[\log q_\phi(a|s_T, m)]$ by summing predicted log probabilities across all joints of a morphology and averaging across all data. We again refer the reader to Appendix A for more mathematical justification.

## 3.3 Task-Agnostic Morphology Evolution (TAME)

TAME combines our task agnostic objective with a basic evolutionary loop similar to that of Wang et al. (2019) to iteratively improve morphology designs from random initialization. While one

might immediately gravitate towards gradient-based approaches, note that arbitrary morphology representations are discrete: the choice to add a limb does not have a computable gradient. We thus turn to evolutionary optimization over a population of morphologies $\mathcal{P}$ each with fitness $f$ equal to an estimate of the objective in Equation 1 computed using $q_\phi$. For each of $N$ subsequent generations, we do the following:

**1. Mutation.** If $\mathcal{P}$ is empty, we initialize it with randomly sampled morphologies. Otherwise, we sample top morphologies according to the current estimate of the fitness function $f$. We then mutate these morphologies by randomly adding or removing limbs and perturbing the parameters of some exhisting limbs and joints using Gaussian noise.

**2. Data Collection.** For each of the newly mutated morphologies, we collect a number of short episodes by sampling actions for each joint $a = (a_1, ..., a_{k(m)}) \sim p(A|m)$ and applying them to the agent in the environment for a small number of steps. At the end of each episode, we collect the final state $s_T \sim p(S_T|a, m)$. We then add the data points $(a, s_T, m)$ to the overall data set used to train $q_\phi$, and add the new morphologies to the population $\mathcal{P}$.

**3. Update.** At the end of each generation, we update $q_\phi$ by training on all of the collected data. Then, we recompute the fitness of each morphology through equation 1 by evaluating the updated $q_\phi$ on all of the data collected for the given morphology.

Finally, we return the best morphology from the population $\mathcal{P}$ according the fitnesses $f$. Our entire algorithm is summarized in Algorithm 1 and depicted in Figure 2. In practice, there are number of additional considerations regarding fitness estimation. In order to get sufficient learning signal, the fitness metric $f(m)$ must be varied enough across different morphologies. For example, if $q_\phi$ were able to predict all actions perfectly, all morphologies would have similar fitness. Practically, we want only the best morphology to be able to achieve high prediction accuracy. To this end we additionally perturb the final states $s_T$ with random Gaussian noise or by adding environment randomization where possible. This also forces morphologies to visit more states, as in the presence of noise, actions will need to result in more distant states to be distinguishable.

TAME offers a number of benefits. First, we never run any type of policy learning, preventing overfitting to a single task and avoiding the slowest step in simulated environments. This makes our approach both more general and faster than task-specific morphology optimization methods. Second, our method is inherently "off-policy" with respect to morphology, meaning that unlike other evolutionary approaches, we do not need to remove bad solutions from the population $\mathcal{P}$. We maintain all of the data collected and use it to train a better classifier $q_\phi$.

**Algorithm 1:** TAME

Init. $q_\phi(a_j|s_T, m)$ and population $\mathcal{P}$;
**for** $i = 1, 2, ..., N$ **do**
    **for** $j = 1, 2..., L$ **do**
        $m \leftarrow$ mutation from $\mathcal{P}$;
        **for** $k = 1, 2, ..., E$ **do**
            sample joint actions $a$;
            $s_T \leftarrow$ endState$(a, m)$;
        $\mathcal{P} \leftarrow \mathcal{P} \cup \{(m, -\infty)\}$;
    $\phi \leftarrow$ train$(q_\phi, \mathcal{P})$;
    **for** $(m, f)$ *in* $\mathcal{P}$ **do**
        $f \leftarrow$ update via equation 1;
**return** $\arg\max_{(m,f)\in\mathcal{P}} f$

*Figure 2.* A visual illustration of the TAME algorithm.

## 4 EXPERIMENTS

In this section, we discuss empirical results from using TAME to evolve agents in multi-task environments. Note that since there are no standard multi-task morphology optimization benchmarks, we created our own morphology representation and set of multi-task environments that will be publicly released. We seek to answer the following questions: Is TAME able to discover good morphologies?; In multi-task settings, how does TAME compare to task supervised algorithms in both performance

and efficiency?; Is TAME's information theoretic objective capable of identifying high-performing morphologies?; What design choices are critical to TAME's performance?; How important are the action primitives?

## 4.1 ENVIRONMENT DETAILS

We evaluate TAME on five distinct environments using the MuJoCo simulator (Todorov et al., 2012), each with a set of evaluation tasks. We represent morphology as a tree structure of "capsule" objects with hinge joints.

- *2D Locomotion*: In the 2D locomotion environment, agents are limited to movement in the X-Z plane. The state $s_T$ is given by the final $(x, z)$ positions of the morphology joints. We evaluate morphologies on three tasks: running forwards, running backwards, and jumping. During evolution we introduce a terrain height-map sampled from a mixture of Gaussians.
- *Cheetah-Like Locomotion*: This is the same as the 2D locomotion environment, but starting with an agent similar to the that of the Half Cheetah from DM Control (Tassa et al., 2018). In the cheetah-like environment only, we fix the limb structure of the agent and use multi-layer perceptrons for $q_\phi$ instead of graph convolutions.
- *3D Locomotion*: In the 3D environment, agents can move in any direction. The state $s_T$ is given by the the final $(x, y, z)$ positions of the joints. We evaluate morphologies on four movement tasks corresponding to each of the following directions: positive X, negative X, positive Y, and negative Y. We again use terrain sampled from a mixture of Gaussians.
- *Arm Reach*: Arm agents can grow limbs from a fixed point at the center of the X-Y plane. For the reach environment, we assess agents by their ability to move the end of their furthest limb to randomly sampled goals within a 1m x 1.6m box. The state is the $(x, y)$ position of the end effector.
- *Arm Push*: The push enviornment is the same as the reach environment, except we introduce two movable boxes, and set the state to the $(x, y)$ position of each box. The agent is assessed on six tasks corresponding to moving each of the boxes in one of three directions.

In each environment an agents final performance is assessed by the average of its performance on each task. In all locomotion experiments, each joint action primitive is given by a cosine function of one of two possible frequencies ($\frac{\pi}{30}$ or $\frac{\pi}{15}$) and one of two possible phases (0 or $\pi$). In all manipulation experiments, each joint action primitive is given by one of four constant torques ($-1, -0.5, 0.5, 1$). Additional evaluations can be found in Appendix C and additional environment details can be found in Appendix E.

## 4.2 BASELINES

We compare our method to a number of different approaches, both task-supervised and task-agnostic.

- *Random*: For this baseline, we randomly sample a morphology train it on the tasks. For the "Cheetah" finetuning experiment *Random* corresponds to random mutation of the original design.
- *TAMR*: Task-Agnostic Morphology Ranking (TAMR) is a variant of our proposed approach without the evolutionary component. Rather than progressively mutating the best morphologies to attain better ones, we generate one large population of morphologies, and rank them according to the same fitness function used in TAME.
- *NGE-Like*: We implement a variant of the task supervised Neural Graph Evolution (NGE) algorithm from Wang et al. (2019). In NGE, each mutated morphology inherits a copy of the policy network from its parent and finetunes it, allowing policies to train over time rather than from scratch. This is a much stronger baseline than standard policy learning with MLPs as each new agent finetunes rather than learning from scratch with a very limited number of samples. For a fair comparison we use the same GNN architecture as TAME for the policy networks instead of the NerveNet model (Wang et al., 2018) from NGE. In our NGE-like variant we also do not use value function based population pruning. We train agent policies in a multi-task environments by giving them a one-hot encoding of the direction for locomotion and the position of the target for manipulation. Fitness is given by the average episode reward over the course of training.

| Env | TAME | TAMR | Random | NGE-Like* | NGE-Prune* | Human |
|---|---|---|---|---|---|---|
| Cheetah | **1000.0 ± 56** | 954.7 ± 25 | 974.6 ± 42 | **1012.8 ± 54** | **1034.3 ± 52** | 846.9 ± 12 |
| 2D | **700.4 ± 43** | 467.2 ± 36 | 329.6 ± 37 | 649.2 ± 43 | 640.8 ± 23 | 846.9 ± 12 |
| 3D | **533.5 ± 39** | 348.0 ± 38 | 302.3 ± 22 | 475.7 ± 48 | **538.3 ± 72** | N/A |
| Reach | **-0.1 ± 15** | **8.1 ± 10** | −40.7 ± 26 | −117.5 ± 14 | −131.5 ± 17 | 89.1 ± .4 |
| Push | **-295.5 ± 16** | −312.7 ± 15 | −449.1 ± 10 | −364.0 ± 24 | −344.3 ± 13 | −388.8 ± 37 |

*Table 1.* Full results across all environments and baselines. Top performers excluding the human designed baseline are bolded. * indicates that the given method had access to task rewards during optimization. The human designed baselines were a version of the cheetah for 2D locomotion and a version of the Gym reacher for manipulation.

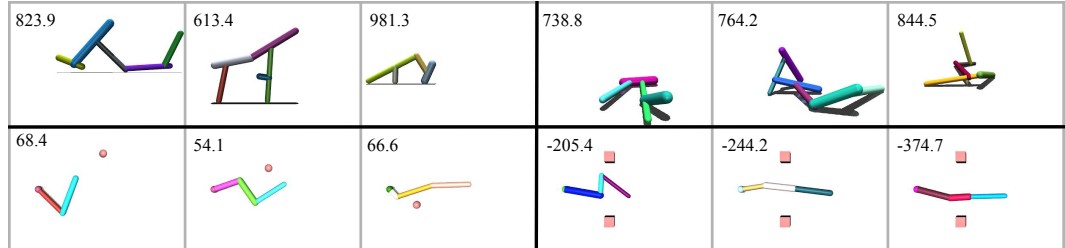

*Figure 3.* Selected evolved agents in each of the four main environments. The top row depicts 2D and then 3D Locomotion and the bottom row depicts Arm Reach and then Arm Push from left to right. At the upper right of each agent is its average performance across tasks. More agent depictions can be found in Appendix F.

- *NGE-Pruning*: This baseline is the same as *NGE-Like*, except with value based population pruning. This is accomplished by training a separate GNN to predict an agent's expected total reward from its morphology embedding. Thompson sampling of the value function (implemented with dropout (Gal & Ghahramani, 2016)) is used to select which mutations to add to the population.

We run all experiments for each environment with the same number of total environment interactions and terrain. For TAME, this amounts to 60 generations of populations of size 24 with 32 short interaction "episodes" for running sampled actions of length 350 for locomotion, 100 for reach, and 125 for pushing. After determining the best morphology, we train it on each of the evaluation tasks with PPO and report the average of the maximum performance across each task. Further experiment details can be found in Appendix G. Each task agnostic method was run for at least twelve seeds. NGE-Like was run for eight seeds.

## 4.3 RESULTS

**Does TAME discover good morphologies?** We plot selected results in Figure 4 and list all results in Table 1. Learning curves for TAME can be found in Appendix D. We observe that within error margins, TAME is a top performer in four of the five environments. Qualitatively, TAME produces 2D hopper-like, walker-like, and worm-like agents, interesting yet effective 3D wiggling agents, and functional arms. Depictions of evolved morphologies can be found in Figure 3. While not consistently ahead of human designed robots across all tasks, in all environments some runs matched or surpassed the performance of the human designed robots. Also, TAME is able to improve upon human designed agents as seen in the Cheetah task.

**How does TAME compare to task supervised methods?** Across all tasks we find that TAME performs similarly to or surpasses the task supervised NGE-like approach for the same number of timesteps. Only in the cheetah-like environment does the task supervised approach do better, likely because of the strong morphological prior for policy learning. Unlike the policy-learning dependent baselines, TAME is able to estimate the quality of a morphology with a very limited number of environment interactions, totaling to around ten or fewer task supervised episodes. When additionally considering that each environment is multi-task, the policy learner in the NGE-like approach is left with only a handful of rollouts with which to optimize each task. This is only exacerbated by the

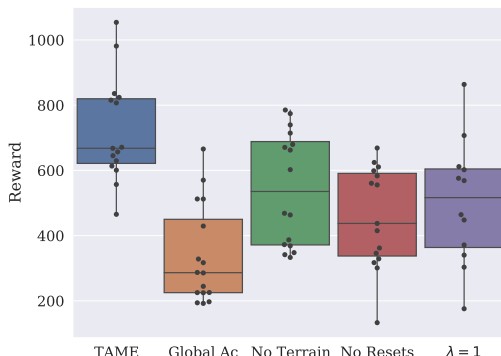

*Figure 4.* A comparison of the performance of TAME versus baselines on four tasks. The vertical line to the right separates the task supervised NGE-like approach. Task-agnostic approaches were run for at least twelve seeds, while NGE-like was run for eight.

challenges of multi-task RL, and the constantly changing dynamics due to morphological mutations. Thus, the NGE-like approach is unlikely to learn an effective policy and provide a strong morphology learning signal. Moreover, NGE only eliminates the bottom portion of the agent population each generation, keeping all but the least performant morphologies around for further policy training. This means that while a run of NGE results in both an optimized morphology and a policy, for the same number of timesteps it evaluates far fewer designs than TAME. NGE-Pruning only shows modest gains over NGE-like most likely because learning an accurate value function with few, noisy evaluations is difficult.

**How fast is TAME?** Table 2 gives the wall-clock performance of different methods in hours. TAME running on two CPU cores is consistently faster than the NGE-Like algorithm running on eight CPU cores. Normalizing for CPU core count, TAME is 6.78x and 5.09x faster on the 2D and Reach environments respectively. As we seek to discover morphologies for harder and harder environments, both policy learning and simulation will become increasingly expensive. By requiring no policy optimization and only a handful of short environment interactions per agent, TAME will be able to more effectively cover a larger morphological search space in a reasonable amount of time.

|  | 2D | 3D | Reach | Push |
|---|---|---|---|---|
| TAME (2) | 2.3 | 3.3 | 1.1 | 1.4 |
| TAMR (2) | 1.2 | 2.6 | 0.4 | 0.6 |
| NGE-Like (8) | 3.9 | 5.2 | 1.4 | 1.8 |

*Table 2.* Wallclock performance comparison in hours. Parenthesis indicate CPU cores used.

| Alg | 2D | 3D | Push |
|---|---|---|---|
| TAME | $700 \pm 42$ | $534 \pm 39$ | $-295 \pm 16$ |
| VarEA | $360 \pm 67$ | $226 \pm 35$ | $-293 \pm 13$ |

*Figure 5.* Selected ablations on the 2D locomotion environment.

*Table 3.* Comparison of TAME and a similar algorithm with final state variance as the objective.

## 4.4 UNDERSTANDING TAME'S PERFORMANCE

In this section, we evaluate various ablations on TAME to understand which aspects of its design are critical to its success.

**Does TAME's objective correlate with morphology performance?** In order assess the quality of our information theoretic objective, we examine two variants of TAME. The first, TAMR, uses the information theoretic objective only to rank a population of agents as discussed earlier. TAMR consistently beats random morphologies in terms of final performance, indicating that our objective is indeed selecting good morphologies. Second, we additionally test the performance of a variant of TAME with a different metric: the variance of final states $s_T$. This has the effect of only measuring fitness by the diversity of states visited and not the controllability of the morphology. Results of

| Primitive Type | 2D Locomotion | 3D Locomotion | Arm Reach | Arm Push |
|---|---|---|---|---|
| Cosine | $700.4 \pm 42.8$ | $533.5 \pm 38.7$ | $-0.1 \pm 14.7$ | $-295.5 \pm 16.1$ |
| Constant | $610.4 \pm 76.6$ | $412.0 \pm 36.8$ | $-36.6 \pm 35.3$ | $-431.7 \pm 17.1$ |

*Table 4.* To test how important selection of the action primitives is in TAME, we compare the final performance of morphologies evolved using different action primitives.

this method on the 2D, 3D, and Push environment can be found in Table 3. In the locomotion environments, this alternative objective fails to create good morphologies.

**What design choices in TAME are important?** In Figure 5 we provide a number of additional ablations on the specific design choices of TAME that increase the quality of the learned fitness function. First, we try sampling the same action primitive for every joint in a morphology. Evidently, by reducing the vocabulary of possible actions, we are unable to get a true sense of how the morphology acts and consequently diminish the accuracy of our fitness metric. This results in much lower performance, approaching that of a random agent. Next, we try removing terrain from the 2D locomotion enviornment, effectively reducing the noise in $s_T$. By making actions easier to classify, the spread of fitnesses is reduced, making it harder for evolution to have a strong signal. In TAME, we reset $q_\phi$ every 12 generations to prevent over-fitting to older morphologies that have been in the dataset for a large number of epochs. This follows similar logic to replay buffer sampling techniques in off-policy RL (Schaul et al., 2015) or population pruning in regularized evolution (Real et al., 2019). By removing resets, we bias the fitness metric towards older morphologies as $q_\phi$ will be better at predicting them. We also analyze the effect of the regularizer $\lambda$ by setting it to one. As expected, this degrades performance as high fitness can be attained by simply adding more joints at the expense of predictability, resulting in agents that are difficult to control. Finally, the lower performance of TAMR in comparison to TAME across all but the reach task as seen in Table 1 indicate that evolution is indeed crucial for finding better solutions.

**How important is the choice of action primitives?** While TAME does not require task specification, it does require specification of a set of per-joint action primitives to use in morphology evaluation. To investigate the importance of choosing apt primitives for a given environment, we swap the joint action primitives between the locomotion and manipulation environments, using constant torques for locomotion and sinusoidal signals for manipulation (Table 4). In the locomotion environments, swapping the primitives results in only a modest degredation of performance, and TAME remains competitive with supervised methods. Given that sinusoidal signals are unlikely to appreciably move a robot arm, manipulation agents suffered a larger performance hit. In general, we find that selecting actions primitives can be easily done by visualizing how randomly sampled morphologies behave in the environment under different action sequences. However, there are scenarios where TAME may be less performant due to its use of action primitives. If one component of the state cannot be effected with the chosen vocabulary of action primitives, TAME may not consider it during optimization. TAME also assumes that the starting state is the same for every episode and agents. In conjunction with relatively short rollouts, this makes it unlikely that TAME will evolve morphologies that can manipulate components of the state space only present far away from initialization. Despite the use of simple action primitives, morphologies evolved by TAME are general enough to complete multiple tasks in both reaching and locomotion settings. We include additional evaluations on a combined reach and locomotion task in Appendix C.

## 5 CONCLUSION

We introduce TAME, a method for evolving morphologies without any reward specification or policy learning. We show that despite not having access to any explicit behavior signals, TAME is still able to discover morphologies with competitive performance in multi-task settings. As TAME only uses randomly sampled action primitives to estimate an information theoretic objective, it is markedly faster than task supervised approaches. Though simple and fast, the use of randomly sampled primitives leaves many open possibilities to be explored in learning morphology through other forms of exploration.

ACKNOWLEDGMENTS

We thank AWS for computing resources. We also gratefully acknowledge the support from Berkeley DeepDrive, NSF, and the ONR Pecase award. Finally, we would like to thank Stas Tiomkin and the rest of the Robot Learning Lab community for their insightful comments and suggestions.

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

## A    EXTENDED DERIVATIONS

Let $S_T$, $A$, and $M$ be random variables representing terminal state, action primitive, and morphology respectively. Note that $A$ is a discrete random variable. We assume morphology $m$ is comprised of $k^{(m)}$ joints and that the original action primitive space $A$ is symmetric for each joint. Thus, the overall primitive action space $A$ for morphology $m$ can be decomposed into $k^{(}m)$ identical primitive spaces $A_j$ for each joint $j$. Mathematically, $A^{(m)} = (A_1, ..., A_{k^{(m)}})$. $|A_j|$ gives the size of the action primitive space for joint $j$. Assume the distribution of each joint action primitives $A_j$ is independent and uniformly distributed. The overall action distribution for a morphology by $p(A|M = m)$. We now provide the full derivation of the objective from Equation 1.

$$\arg\max_m I(S_T; A|M = m)$$

$$= \arg\max_m H(A|M = m) - H(A|S_T, M = m)$$

$$= \arg\max_m H(A|M = m) + \mathbb{E}_{a \sim p(A|m), s_T \sim p(S_T|a,m)}[\log p(a|s_T, m)]$$

$$\geq \arg\max_m H(A|M = m) + \mathbb{E}_{a \sim p(A|m), s_T \sim p(S_T|a,m)}[\log q_\phi(a|s_T, m)]$$

$$= \arg\max_m \log |A_j|^{k^{(m)}} + \mathbb{E}_{a \sim p(A|m), s_T \sim p(S_T|a,m)}[\log q_\phi(a|s_T, m)]$$

$$= \arg\max_m k^{(m)} \left( \log |A_j| + \frac{1}{k^{(m)}} \mathbb{E}_{a \sim p(A|m), s_T \sim p(S_T|a,m)}[\log q_\phi(a|s_T, M = m)] \right)$$

$$\geq \arg\max_m (k^{(m)})^\lambda \left( \log |A_j| + \frac{1}{k^{(m)}} \mathbb{E}_{a \sim p(A|m), s_T \sim p(S_T|a,m)}[\log q_\phi(a|s_T, M = m)] \right)$$

Where $0 \leq \lambda \leq 1$ is a regularizer on the effect of adding more joints. We can additionally derive a per-joint variant of the objective by assuming that the action primitive taken by each joint is conditionally independent given the state and morphology, or $p(a|s_T, m) = p(a_1|s_T, m) \cdots p(a_{k^{(m)}}|s_T, m)$. This allows us to use the classification per-joint from $q_\phi$.

$$\arg\max_m (k^{(m)})^\lambda \left( \log |A_j| + \frac{1}{k^{(m)}} \mathbb{E}_{a \sim p(A|m), s_T \sim p(S_T|a,m)}[\log q_\phi(a|s_T, M = m)] \right)$$

$$= \arg\max_m (k^{(m)})^\lambda \left( \log |A_j| + \frac{1}{k^{(m)}} \mathbb{E}_{a \sim p(A|m), s_T \sim p(S_T|a,m)} \left[ \log \prod_{j=1}^{k^{(m)}} q_\phi(a_j|s_T, m) \right] \right)$$

$$= \arg\max_m (k^{(m)})^\lambda \left( \log |A_j| + \frac{1}{k^{(m)}} \mathbb{E}_{a \sim p(A|m), s_T \sim p(S_T|a,m)} \left[ \sum_{j=1}^{k^{(m)}} \log q_\phi(a_j|s_T, m) \right] \right)$$

$$= \arg\max_m (k^{(m)})^\lambda \left( \log |A_j| + \frac{1}{k^{(m)}} \sum_{j=1}^{k^{(m)}} \mathbb{E}_{a \sim p(A|m), s_T \sim p(S_T|a,m)} [\log q_\phi(a_j|s_T, m)] \right)$$

The final average and expectation term can be interpreted as the negative cross entropy loss of $q_\phi$ on joints from morphology $m$. This is how the value is practically computed in our experiment code.

## B    RESOURCES

As mentioned in the abstract, our morphology evolution code and videos of our learned agents can be found at https://sites.google.com/view/task-agnostic-evolution.

## C    ADDITIONAL EVALUATIONS

We additionally evaluated TAME and our baselines on more complex tasks for the 2D locomotion and 3D locomotion environments to demonstrate learned morphologies are adaptable to more scenarios.

| Method | TAME | TAMR | Random | NGE-Like* | NGE-Pruning* |
|---|---|---|---|---|---|
| 2D Locomotion | $113.8 \pm 15.9$ | $75.3 \pm 20.1$ | $24.2 \pm 6.6$ | $123.8 \pm 21.9$ | $152.0 \pm 21.1$ |
| 3D Locomotion | $7.4 \pm 1.2$ | $7.7 \pm 1.5$ | $3.7 \pm 1.1$ | $4.2 \pm 1.4$ | $5.3 \pm 1.4$ |

*Table 5.* Additional evaluation of methods on a goal reaching task in the 2D and 3D environments. * denotes having access to task specification and rewards during morphology optimization.

Rather than just moving in a pre-specified direction, we evaluate agents on their ability to reach goals. In 2D the goals are -8m, -4m, 4m, and 8m along the $x$-axis. In 3D the goals are given by the $x$-$y$ positions $(-4, 0), (0, -4), (4, 0), (0, 4)$ in meters. The reward is given by the decrease in $l2$ distance before and after executing a given action, and a $+1$ bonus for being within 0.1m of the goal. We added the goal position minus the current root position to the state space to tell the agent where the goal is. In our evaluation we used the same agents for TAME, TAMR, and Random, but re-ran morphology optimization for the NGE-like and NGE-Pruning methods as the reward function changed. Results are given in Table 4, showing the average reward across policies trained independently for each goal via one million steps of PPO, unlike in other locomotion experiments which were run for 500,000 steps. We again find that TAME is within confidence bounds of the regular NGE-like method despite not having access to task specification or rewards, though it is slightly outperformed by the NGE-Pruning method. In 3D, TAME does better than the task-supervised methods, likely because policy learning is much more difficult in 3D resulting in worse morphology signal for the NGE style baselines. Additionally, we investigate the importance of reward and state space specification for the NGE-like methods. Rather than providing the difference between the goal position and the current position to the agent, we re-ran the 2D experiments but instead provided a one-hot encoding of the goal and the absolute position of the agent. In this setting, performance on the NGE baselines dropped to $72.8 \pm 24.6$ and $95.0 \pm 20.5$ for NGE-like and NGE-Pruning respectively. This is likely because it is harder to re-use information across tasks when the goal is provided as a one-hot vector instead of relative position.

## D  LEARNING CURVES

Here we provide the learning curves for TAME.

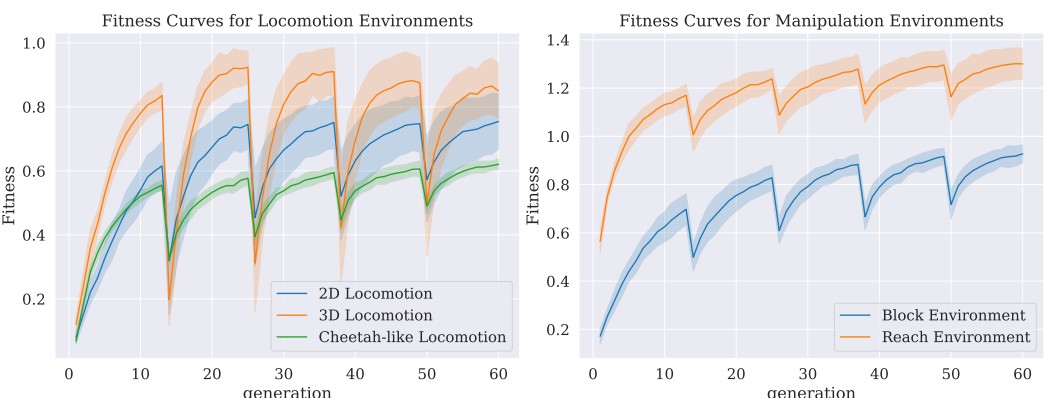

*Figure 6.* Learning curves plotting the TAME information theoretic objective during evolution. Error bars are the standard deviation across seeds. Note that $q_\phi$ was reset every 12 generations.

## E  ENVIRONMENT DETAILS

Here we provide additional specification on the environments used to assess TAME. Images of the environments used during evolution can be found in Figure 7.

**Morphology Representation.** We implement morphologies using tree like objects and MJCF in Deepmind's DM Control Suite (Tassa et al., 2018). Morphologies are composed of nodes which all

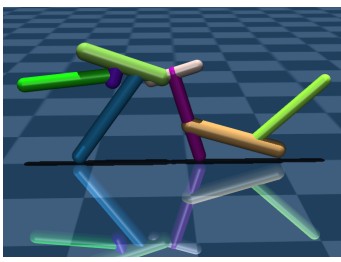 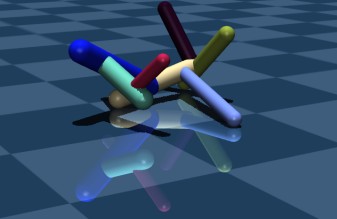 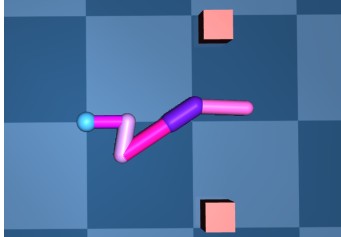

*(a) A random morphology in the 2D environment*  *(b) A random morphology in the 3D environment*  *(c) A random morphology in the block environment.*

*Figure 7.* Images of all environments used in experiments.

take the form of "capsule" objects. To fully specify a node, one must provide its radius, x-length, y-length, and z-length. In 2D and Arm environments, we restrict lengths in some directions to be zero. Nodes are connected to one another using fixed or hinge joints. Given a morphology is represented by a tree, a connection between nodes will always have a parent and a child. A joint is fully specified by a type (fixed or hinge on relative z or y axis of the parent), an attachment point from zero to one that specifies how far down the extent of the parent node the child node attaches, a symmetric joint range, and a joint strength or gear.

An additional consideration with this arbitrary morphology representations are collisions. Without collision detection, limbs would be allowed to grow into each other, resulting in unrealistic morphologies and unstable simulations. While previous works that use MuJoCo for morphology optimization circumvent this problem by disabling contacts between the limbs of morphologies (Wang et al., 2019), we develop a contact detection system that prevents random sampling or mutations from creating invalid structures. As such, all of our environments have collisions between limbs enabled.

**Morphology Sampling.** Random morphologies are recursively sampled in BFS order. First, we randomly generate a root node by sampling parameters uniformly over allowed ranges and add it to a node queue. While the node queue is non-empty and we are under the maximum allowed number of limbs, we pop a parent node from the queue, run a Bernoulli trial to determine if it will grow a child node, and enqueue the randomly sampled child node if created. In order to allow for more uniform sampling of designs, for sampling arm agents we first sample a number of limbs uniformly at random, and then repeatedly mutate the agent until it has that number of nodes.

**Morphology Mutation.** We mutate morphologies in three steps. First, we iterate over all of the nodes of a morphology and with some probability perturb its parameters with Gaussian noise. Note that we consider the edge or joint attributes to be part of the child, and perturb these parameters with those of the child's limb. Next, we collect all the nodes that have not maxed out their child count, and have each of them grow a child node with some small probability. Finally, we collect all the leaf nodes of the tree, and delete each of them with some small probability.

**Locomotion Details.** In locomotion environments agents are tasked with moving in different directions. We use physics parameters similar to those of the DM Control cheetah, though we remove the total-mass specification. We restrict capsules to be at most length 0.75m in each direction and have a radius between 0.035m and 0.07m. Joints are restricted to having a range of 30 and 70 degrees in either direction with a gear between 50 and 100. In the 2D environments, we restrict the agent to only have limbs grow in the X and Z directions and have joints on the Y-axis. Each node in the morphology tree is allowed to have at most two children, and during random generation children are sampled with probability 0.3. Parameters of each node are perturbed with probability of 0.14 for the geometric attributes and 0.07 for the joint attributes. During evolution, we use a standard deviation of 0.125 in modifying the lengths of the nodes, add a limb with probability 0.1, and remove a limb with probability 0.08. We also force all edges to be joints and do not allow static connections, except in the cheetah environment.

The reward for the different movement tasks in the X, negative X, Y, and negative Y directions is given by $\min(10, \max(-10, v))$, where $v$ is the agent's linear velocity in the desired direction. For

the Z movement task, the reward is given by $\min(10, \max(-2, v))$. In the main body of the paper, we refer to X, negative X, and Z as forward, backward, and jumping respectively.

Finally, we sample terrain height maps using random Gaussian mixtures as depicted in Figure 8. For the 2D environments, the maximum terrain height is 0.2m, while for the 3D environments, it is 0.25. However, the variance for the sampled 2D Gaussians is restricted to be between 0.2m and 0.8m, making the 2D terrain bumpier than the 3D terrain that results from randomly sampled covariance matrices. For locomotion tasks we additionally introduce Gaussian state noise of mean zero and standard deviation 0.05.

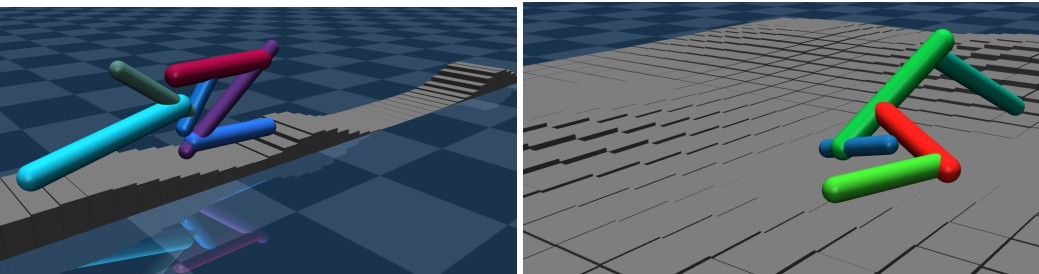

*Figure 8.* Images of random agents in 2D and 3D terrain respectively.

**Manipulation Details** In manipulation environments, arm-like agents either reach goals or attempt to push blocks to specified locations. For these environments, we use similar parameters to that of the Open AI Gym reacher (Brockman et al., 2016), though with scaled up bodies. During the construction phase, we limit the limbs of arm-agents to only grow in the X direction to a maximum length of 0.75 and radius between 0.04m and 0.07m. The base of the arm is fixed at (0,0). Joints between limbs of the arm are restricted to the Z-axis and have ranges between 90 and 175 degrees in either direction and gearings between 70 and 80. Nodes in the arm environment can only have one child. They grow or remove children with probability 0.2. The perturbation probabilities are the same as in locomotion.

The reward for the reach task is given by the negative l2 distance between the end of the agent's further limb and the pre-selected goal position from a box of size 1m by 1.6m. A bonus of 100 is given when the agent is within 0.1m of the desired position. The push task is more complicated. Two 0.2m boxes, box 1 and box 2, are introduced at (0.9, 0.65) and (0.9, 0-0.65) respectively. There are six total pushing tasks that result from moving each of the boxes to one of three positions. Box 1 is pushed to (0.0, 0.65), (0.9, 1.4), and (0.15, 1.25). Box 2 is pushed to (0.0, -0.65), (0.9, -1.4), and (0.15, -1.25). The reward for the box pushing task is given by $-||\text{box} - \text{target}||_2 - 0.5||\text{box} - \text{arm}||_2$.

For manipulation tasks we additionally introduce Gaussian state noise of mean zero and standard deviation 0.2.

# F  ADDITIONAL MORPHOLOGY RESULTS

Here we provide images of additional morphologies constructed using our algorithms and baselines.

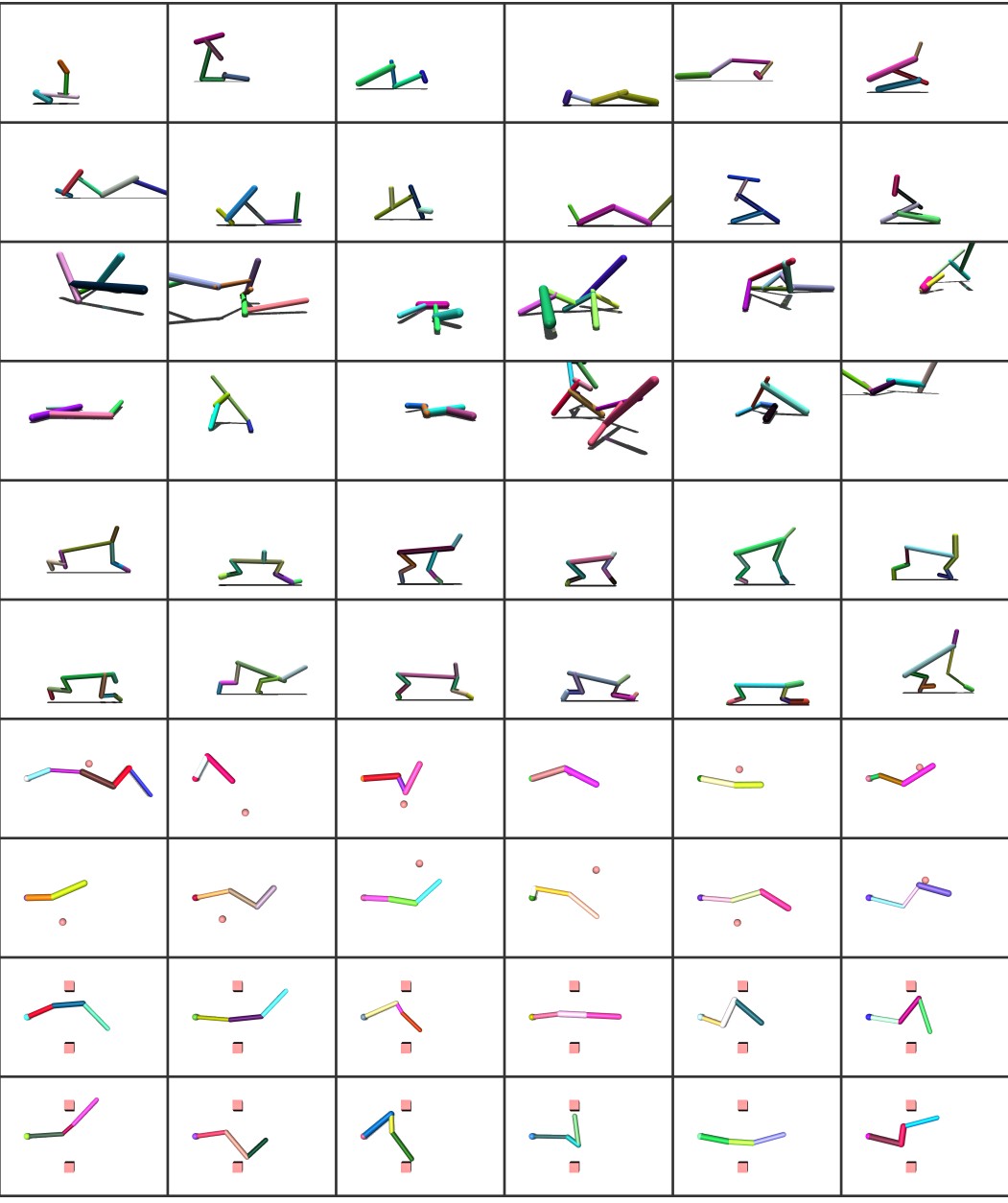

*Figure 9.* Morphologies evolved by TAME on all tasks.

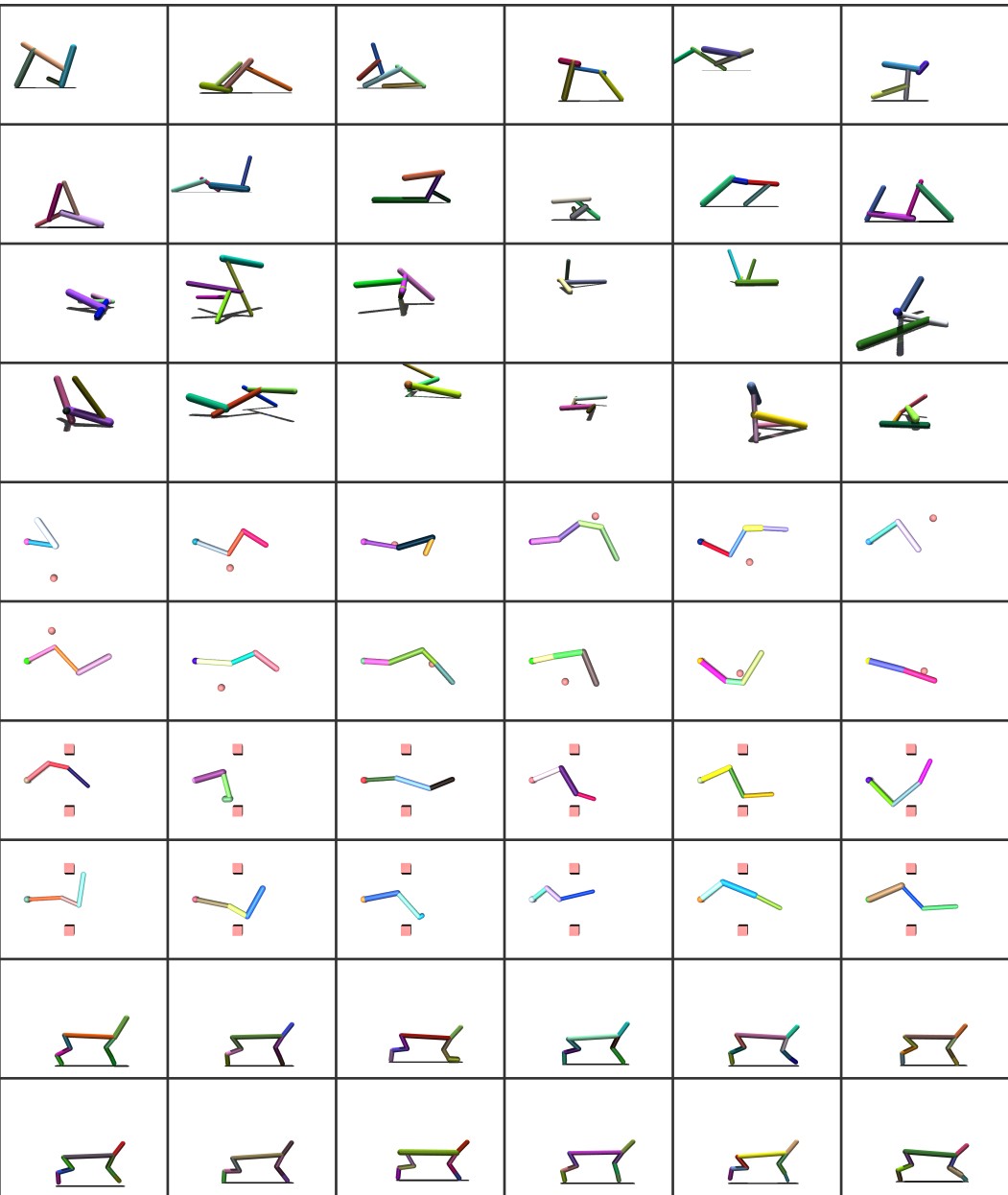

*Figure 10.* Morphologies evolved by TAMR on all tasks.

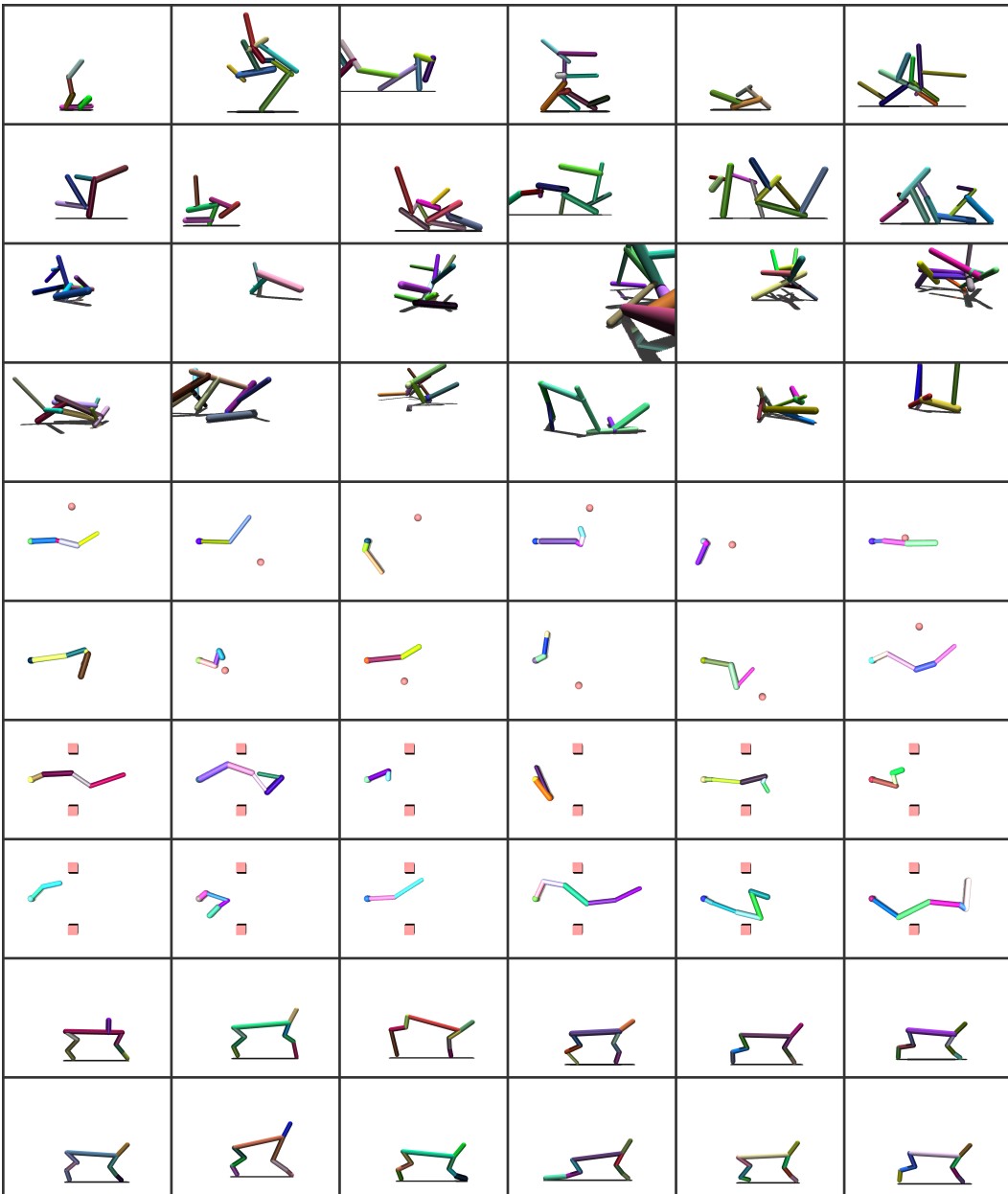

*Figure 11.* Randomly sampled morphologies for each task.

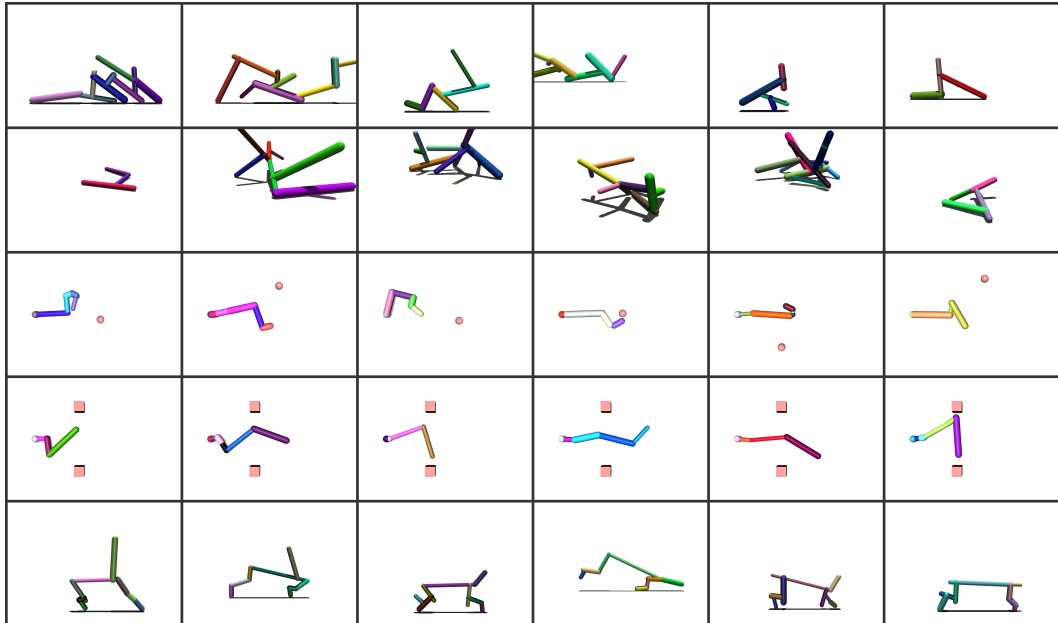

*Figure 12.* Morphologies evolved by the NGE-Like algorithm on all tasks.

## G  ARCHITECTURE AND HYPERPARAMETERS

We implement all graph models using Graph Convolutions from the Pytorch Geometric software package (Fey & Lenssen, 2019). Graph convolutions are able to develop global features by passing messages between nodes. Each node in the graph computes a message via a fully connected layer and a state using a different fully connected layer. Next, each node averages all of its incoming messages and adds it to its state. For more detail, we refer the reader to Morris et al. (2019). All of our graph networks for both TAME and NGE-like have three Graph Convolutions and use ReLU activations. For policy evaluations, we use the PPO implementation from Stable-Baslines 3 (Raffin et al., 2019).

Below we provide list all hyper-parameters. Note that all experiments were run for the same total number of timesteps as mentioned in Section 4. Thus, for the NGE-like baseline, each individual ran policy learning for $32 \times 350 = 1120$ timesteps per generation in the locomotion tasks. The same relationship holds for manipulation. Table 6 lists hyperparameters used for evaluations. Table 7 lists hyperparameters used for evolution.

| Hyperparameter | Value |
|---|---|
| Discount | 0.99 |
| GAE Parameter | 0.95 |
| steps per udpate | 1000 |
| epochs per update | 8 |
| batch size | 128 |
| learning rate | 0.0003 |
| # workers | 1 |
| entropy coef | 0.001 |
| Hidden Layers | 2xDense256 |

*Table 6.* PPO Hyperparameters used for all policy evaluations. For each locomotion task, we trained policies for 500k timesteps per task. For the arm reach task, we trained policies for one million timesteps, and for the push task we trained policies for 200k timesteps per task.

| Hyperparameter | Value |
|---|---|
| generations | 60 |
| population size | 24 |
| episodes/individual | 32 |
| learning rate | 0.001 |
| batch size | 128 |
| epochs/gen | 15 |
| lambda | 0.25, 0.2 |
| hidden layers | 3xGraphConv192 |
| reset freq | 12 |
| prediction classes | 4 |
| mutate % | 0.06 |

*Table 7.* Evolution hyperparameters. If two values are listed, the first set are for locomotion environments, the second are for manipulation. In practice we additionally take the natural log of $k^{(m)}$. Mutate % refers to the percentage of the population we randomly sample morphologies from. For NGE-Like, we maintain 60% of the population each generation, and sample a new 40% from the existing population.

