# OpenReview forum: "Task-Agnostic Morphology Evolution"
_ICLR.cc/2021/Conference — ICLR 2021 Poster_

### Official Review · AnonReviewer4 · 2020-10-28
**review for "Task-Agnostic Morphology Evolution"**

**Rating:** 6
**Confidence:** 3

**Review:**

The paper introduces an algorithm for optimizing the robot morphology in a simulated environment. The key idea is that instead of finding a morphology and a controller for a specific task, they propose to search for a morphology that can reach a large variety of states in a predictable way. Specifically, they developed an objective function that maximizes the mutual information between the actions and the final states of the robot. The proposed algorithm is demonstrated on a few simulated locomotion and manipulation tasks.

The paper is well written and the problem that the work is addressing is significant in my point of view.

What I like about the work:
1. The idea of optimizing morphology in a task-agnostic way is interesting.
2. The proposed formulation and algorithm seems concrete and reasonable.
3. The proposed algorithm seems effective for the designed experiments presented and compares favorably to baseline methods.

My concerns about the work:
1. The proposed objective only considers reachability of the robot, but not the efficiency of completing the task. A good baseline would be whether the optimized design works better than some human-designed morphology. For example, does the optimized result outperform the hopper, halfcheetah, and the 2d manipulator from OpenAI gym environments?
2. Is the algorithm sensitive to the choice of the regularizer lambda that balances adding limbs and prediction accuracy?
3. The actions used to explore the environment is quite simplistic (sinusoidal-based for locomotion and constant for manipulation). Though it works for the presented problems, it's not clear if it will work for more complex problems. For example, if we want to design a biped robot that does not fall to the ground, random exploration would likely give us a very narrow range of states.

In general I think the paper introduces an interesting idea. It would be a good contribution to ICLR if my concerns above could be addressed.

---

> ### Author Response · Authors · 2020-11-12
> **Response to Reviewer #4**
>
> We thank the reviewer for characterizing our paper “well written” and  tackling a problem that is “significant”. We have run additional experiments and updated the paper. Below we address the reviewers concerns:
>
> *> 1. The proposed objective only considers reachability of the robot, but not the efficiency of completing the task. … human designed morphology*
>
> While it is impossible to measure the efficiency of a task without any task specification, we argue that our approach does provide a proxy to this. In order to easily predict the action from the terminal state, the terminal states must be diverse. If an agent is unable to efficiently move throughout the state space, all its terminal states will be similar, making it hard to predict the agent's actions resulting in low fitness.
>
> To compare to human designs, we have added the performances of gym-like agents to Table 1. We find that TAME is able to improve the performance of cheetah in the finetune setting above human design levels and that many of the run seeds result in comparable performance to cheetah in the random 2D setting, though the average is slightly lower. TAME outperforms the gym reacher in the push setting, but is outperformed in the reach setting. This matching of performance is consistent with Wang et al., where the evolved arbitrary topology 2D locomotion agent achieves rewards only slightly above that of the DM control cheetah, but with the additional freedom to have its limbs collide while we enforce contacts.
>
> *> 2. Is the algorithm sensitive to the choice of the regularizer lambda*
>
> We have added an additional ablation to study the effect of the choice of lambda, which is now depicted in the ablations figure and discussed in section 4.4. We find that removing the regularization does indeed lower the performance of TAME as the evolutionary algorithm begins to favor adding more, random limbs instead of good, predictable ones.
>
> *> 3. The actions used to explore the environment is quite simplistic… complex problems… bipedal robot.*
>
> We have added an additional ablation study and discussion on the choice of action primitives in section 4.4 along with more discussions of where TAME is applicable.
>
> We believe the evaluation tasks chosen are general enough to be applicable to more settings: for example, multi-directional locomotion is a prerequisite to goal reaching and end-effector goal reaching is a prerequisite to manipulation. Existing work, “Modular Robot Design Synthesis with DeepRL”, by Whitman et al. even use reachability as a supervised objective for morphology optimization.
>
> For the more complex problems on the axis of morphology discussed in the review, TAME could be relevant in the fine-tuning setting. As now discussed in section 4.4, one could visualize the movement of variants of a base bipedal morphology under different action sequences to choose primitives for TAME. In the base setting, however, TAME is not intended to solve morphological tasks as it is designed to choose the morphology. Without constraints, it will be likely to choose whatever it finds most efficient. This is also a limitation of supervised approaches and our baselines -- learning to crawl is often easier than learning to walk, so unless one were to somehow specify a good reward function for upright bipedal locomotion on arbitrary topologies, task supervised methods would still be likely to learn efficient crawling agents rather than walking ones.
>
> The bipedal example, however, does bring up an additional point. TAME uses short episodes of action primitives to estimate fitness, making it more difficult to discover components of the state space only introduced far away from initialization. We have added a discussion of this limitation at the end of section 4.4.
>
> Thank you for your review and please let us know if you have any more questions or requested updates.

---

> > ### Comment · AnonReviewer4 · 2020-11-24
> > **follow-up**
> >
> > Thank you for your detailed response!
> >
> > The additional experiments and ablations are quite helpful in addressing my concerns. I also appreciate that the authors discussed the limitations of the methods in handling more complex scenarios directly. As such I would increased my score to 6.
> >
> > My main remaining concern is the comparison to human-made designs. From the provided study, it seems that human design outperformed the optimized design in 2 out of 4 cases. Furthermore, in the cheetah case the human design is even worse than the randomly picked design. I'm wondering how many random designs are tested for it? Is it because it happens to pick a good one?

---

### Official Review · AnonReviewer2 · 2020-10-28
**Review of TAME**

**Rating:** 7
**Confidence:** 4

**Review:**

Summary:

This paper develops a general morphology evolution algorithm, and demonstrates its utility in a setting where morphologies are encoded as graphs. The methodology is grounded in a theoretical notion of empowerment, and theory is introduced that extends empowerment to the case of morphology. The morphology evolution itself requires no task-specific signals, but yields morphologies that generalize well to several tasks of interest.


Strong points:

The extension of empowerment to morphologies is novel, interesting, and well-motivated.

The use of random actions is satisfying in that it is initially counter-intuitive, but well-grounded, and in line with other recent advances based in generating interesting things from random input (e.g., GANs).

The introduced space of morphologies is quite general and can capture some very interesting designs.

The ablations in Figure 5 and Table 3 are very helpful in validating the theory. Table 3 in particular verifies how empowerment can provide an advantage over more naiive diversity metrics.




Weak points:

There are some key gaps in related work literature.

One is NS-LC (Evolving a Diversity of Creatures through Novelty Search and Local Competition,), which was used to evolve explicitly diverse high-performing morphologies (and is an example of the broader category of quality diversity algorithms). The goal of that work is different: to collect diverse morphologies, instead of finding one that is maximally general, but the methodology is related. There is generally a lack of references in the description of the evolutionary algorithm (EA) used in the paper. Is this similar to algorithms used to evolve other sorts of open-ended structures? E.g., open-ended neural network design (e.g., Evolving Deep Neural Networks)? Or EAs that include generation-by-generation updates of a model like the classifier used in this paper? E.g., surrogate-assisted methods?

There is also a missing link to recent advances in self-supervised methods.

“we reset q_phi every 12 generations to prevent over-fitting to older morphologies” Worth mentioning the connection of this to ideas like experience replay in RL and removing the oldest individual in Regularized Evolution.

The theory motivates the use of random actions, but the experiments use a highly restricted set of actions that are well-suited to the eventual tasks of interest, which takes away from the promise of generality and task-agnosticism of the method. Should we expect these specialized “predefined action sequences” to generalize to a wide variety of morphologies and tasks, or does the experimenter need to design these by hand depending on the tasks?

“each joint action is given by a cosine function of one of two possible frequencies and one of two possible phases”. What are these frequencies and phases? So, each joint has 4 possible controllers. Are combinations of joints with these controllers sufficient for performing arbitrary tasks?

Similarly, the tasks in the paper are also closely-aligned with the empowerment objective. That is, the state space is low-dimensional, and task is measured in a 1- or 2-dimensional slice of the state space. Can we expect the approach could generalize to more complex tasks? E.g., can the approach handle a task that combines locomotion and reach, i.e., navigate to randomly sampled goals when the agent is not tethered? Why not evolve a single morphology that can solve all the classes of tasks in experiments (except cheetah)? Isn’t this the promise of the method?

The changes from NGE to NGE-like make the method more comparable to TAME, but the paper should elaborate on what the missing parts do and whether TAME could take advantage of them. E.g., what is “value based population pruning” and why doesn’t TAME use it? What is NerveNet and why doesn’t TAME use it? Why was NGE-like run for 8 seeds instead of 12? Because it’s slower? Why are NGE-like and TAME run on different numbers of cores?

One key baseline missing is single-task or multi-task version of the EA that TAME uses, but simply using task fitness instead of empowerment. This would give a sense of how the task-specific variant of your underlying EA compares to NGE, and validate whether the EA is at least powerful enough to take advantage of task signals when they are available.



Minor comments:

“Additionally, morphologies with better exploration will be able to learn to solve tasks more quickly.” Is there existing work that shows this or some other explanation? It makes intuitive sense, but may not be true in general.

In Eq. 1, looks like “log|J|” should be “log|A_j|”, and the expectation “E” should be formatted as in the previous lines.

Should TAME Cheetah be bold in Table 1 since it is within the margin of NGE-like?

In “How does Tame compare to task supervised methods?”: “We hypothesize that this” -> Clarify what “this” is and why NGE-like is good on Cheetah (because the morphology space is more limited or structured?).

-------
Based on the author response, the rating has been updated (see comments below).

---

> ### Author Response · Authors · 2020-11-12
> **Response to Reviewer #2 (Part 1/2)**
>
> We thank the reviewer for finding our work “novel, interesting, and well-motivated”. We hope to address any concerns with the following changes and additional experiments now included in the paper. We are happy to discuss any further questions.
>
> *> NS-LC*
>
> We have subsequently added a discussion NS-LC and other QD approaches to the related work section. QD approaches like NS-LC still require a task based fitness function, and thus we believe our approach to be complementary to, not competitive with, QD approaches as the TAME objective could be substituted for task rewards to evolve a diverse population.
>
> *> lack of references in the description of the evolutionary algorithm (EA) used in the paper*
>
> We use the most basic evolutionary approach of 1. Initialize a population 2. Compute fitness 3. Mutate 4. Repeat 2-3 until convergence. This basic evolutionary loop is similar to that of NGE and we have appropriately made reference to this in section 3.3. One critical difference between our approach and NGE is that we do not eliminate individuals from the population and continually use their data to update q_phi. We provide a more in-depth discussion of this in our later comments.
>
> *> There is also a missing link to recent advances in self-supervised methods.*
>
> We have now added references to contrastive mutual information style methods to the related work in addition to the existing references for self-supervised skill discovery work (DIAYN, DADS).
>
> *> we reset q_phi every 12 generations*
>
> The method section has been updated to include the listed connections and relevant citations.
>
> *> the experiments use a highly restricted set of actions that are well-suited*
>
> We ran additional ablations to study this exact issue and discuss them under a new heading in section 4.4. Empirically, we find that even using a different primitive space our method is still competitive to the NGE-like baselines in the locomotion environments. The action sequences do encode a bias towards the type of environment the agents are in, but not necessarily the task. However, the primitives are a) easy to select from the environment before task specification and b) aren’t the largest factor in performance.
>
> *> What are these frequencies and phases?*
>
> We have added these additional experiment details to section 4.1
>
> *> Are combinations of joints with these controllers sufficient for performing arbitrary tasks?*
>
> In general, we find that random controllers based on the pre-defined primitives are not enough to complete the task. While some agents can demonstrate forward / backward motion, most do not perform the task close to as well as with the later tuned policy. The reach task, for example, is unable to be completed with a constant torque to each joint. This is also particularly evident in the 3D environment. We encourage the reviewer to examine our video of results (https://sites.google.com/view/task-agnostic-evolution) starting at 3:18 where we show what agents look like using the random controllers versus the final trained policies.
>
> *> an the approach handle a task that combines locomotion and reach?... Why not evolve a single morphology that can solve all the classes of tasks in experiments*
>
> We are currently running experiments on locomotion and reach. We believe this would be an easy scenario for the learned morphologies (particularly those in 2D) to handle as they can already locomote and are not biased learning locomotion in one direction during training. We have added additional discussion about the applicability of TAME at the end of section 4.4.
>
> In our survey of literature, we found that tasks used for free topology morphology optimization were generally simple given the high-dimensional complex nature of the problem. For example,  Cheney et al 2013, Wang et al. 2019, Shaff et al 2019, Ha et al 2019, and Luck et al 2020 all use forward locomotion as the primary task. By optimizing for the multi-task case we believe we are breaching a new barrier. In designing our evaluation tasks we considered what would be desired out of agents in the given environment. While perhaps we would want agents in a plane to push boxes, its more likely that we care about their ability to navigate.
>
> ... continued in part 2/2.

---

> > ### Author Response · Authors · 2020-11-12
> > **Response to Reviewer #2 (Part 2/2)**
> >
> > *> changes from NGE to NGE-like make the method more comparable, but the paper should elaborate on what the missing parts do *
> >
> > We have expanded the set of baselines to include a version of NGE-like with value-based pruning. Here are some additional clarifications on NGE:
> > - The pruning method from NGE  learns a GNN (with dropout) that predicts the expected reward of a given morphology using fitness data (a surrogate). Using dropout-based thompson sampling, candidate designs are chosen to add to the population. This has the effect of not wasting compute on bad mutations. Such a surrogate is inapplicable to TAME as TAME requires samples from the environment to estimate the information theoretic objective.
> > - NerveNet is a GNN structure utilized by Wang et al. which was directly carried over to NGE. We did not extend the NerveNet code from NGE because despite the paper being released in 2019, the code was written in tensorflow 1.0.0 (not a newer version of TF like 1.14). Additionally, NerveNet uses internal RNN-like propagation steps making it much slower than our proposed GNN architecture.
> > - Besides the GNN architectures, our baselines should be directly comparable to NGE.
> >
> > *> Why was NGE-like run for 8 seeds instead of 12? Because it’s slower? Why are NGE-like and TAME run on different numbers of cores?*
> >
> > NGE was run for 8 seeds instead of 12 because of computational limitations. We could run far more seeds for TAME because of its speed. NGE requires updating policies for each agent in the population. This process was parallelized across cores. In TAME, we ran everything serially. One could parallelize the data collection process across multiple cores, but given the algorithms existing speed we didn’t believe it was necessary. NGE-like was run on 8 core partitions while TAME was run on two core partitions, though it was really designed to only use one core. If anything, we believe our speed evaluation benefited NGE.
> >
> > *> One key baseline missing is single-task or multi-task version of the EA that TAME uses, but simply using task fitness instead of empowerment.*
> > We intended NGE-like to be this baseline. In the text we now additionally clarify that the EA we use is the same as that of NGE except that we maintain all individuals in the population. The “multi-task version” with the same EA is intended to be NGE-like. Here are the exact differences between the NGE-like baseline and TAME:
> > - NGE-like removes the least fit individuals from the population, TAME maintains all individuals in the population
> > - NGE-like has access to task specific rewards, and uses them to learn task conditioned policies with PPO and a GNN.
> > - NGE-like uses evaluations of the learned policies to estimate fitness, rather than the information theoretic objective.
> > An even weaker baseline would be to use MLP policies instead of GNN-based ones, but this would require re-learning new policies for each different topology, which in the NGE publication (Wang et al.) was found to perform far worse. Please let us know if you have any questions!
> > We have also addressed all minor comments in the paper text.
> >
> > Thank you for your thoughtful review and in-depth review, and let us know if there are any outstanding concerns or desired clarifications as we continue to update the paper.

---

> > > ### Comment · AnonReviewer2 · 2020-11-24
> > > **More confident in the significance of the approach**
> > >
> > > Based on the response from the authors, I have decided to increase my rating of the paper.
> > >
> > > The authors have made clear that the main contribution of the approach is to bring a key strength from evolutionary methods, i.e., rapid adaptation via task-agnosticism, to a problem where simple fitness-based evolution, i.e., NGE, has been shown to be successful. Such rapid adaptation capability will be critical when producing physical robots from evolved designs.
> > >
> > > By presenting a particular task-agnostic approach, they have shown that the area is ripe for incorporation of other evolutionary methods that emphasize adaptation and diversity at a population-level.
> > >
> > > The new experiments, especially the challenging 3D locomotion+reach experiment, demonstrate that this type of approach could enable scaling beyond fitness-based approaches like NGE. And the authors have made clear that, although apparently simple, the level of complexity of this tasks is beyond that of previous work.
> > >
> > > I still think there is a notable lack of background and motivation for why evolution is the right framework for integrating empowerment with automated morphology design. As is, the impression is "NGE does it, so we do it", but there is a deeper connection that could lead to more interesting follow-up work if it were articulated well in this paper.

---

> > > > ### Comment · AnonReviewer2 · 2020-11-24
> > > > **NGE citation should be updated to ICLR**
> > > >
> > > > NGE is the most important reference in the paper; it's citation in the references should point to the ICLR version, not the arxiv version.
> > > >
> > > > Also, when NGE is first cited in the text, the parentheses around the citation are missing.

---

### Official Review · AnonReviewer1 · 2020-10-29
**Interesting research in evolving robot morphologies**

**Rating:** 7
**Confidence:** 4

**Review:**

The paper presents a method to evolve morphologies of a robotic system without explicit reward and using an empowerment-like quantity. The thus obtained morphologies can get higher rewards in task settings when RL algorithms are applied.

The story and presentation of the paper are clear. I also like the results and think it is interesting, maybe more for a conference like ALife than ICLR though.

Strengths:
- Interesting way to estimate the information criterion using a GNN
- formulation of a morphology-empowerment
- analysis and ablations justify the design choices and the method
- good results (on self-given tasks)

Weaknesses:
- the mathematical formulation is sloppy in many places
- the morphologies are compared on different sequences. It might be interesting to know how much variance the estimation of Eq 1 has when sampling a different batch of action-sequences

I think when the problems with the formulation are fixed and the typos are removed the paper can be much stronger.

Related work: As you quantity is very close to the Empowerment definition, I think the original work by the Polani group should be cited, e.g. "Empowerment: A universal agent-centric measure of control", Klyubin, Polani, Nehaniv
An interesting combination could be to use task agnostic live-time adaptation, such as predictive information maximization (Information Driven Self-Organization of Complex Robotic Behaviors, PLoSOne) or related work could be combined.

Problems in the mathematical formulation:
Page 3, first equation (before (1)): What is the expectation taken over?
You write about q_theta being a classifier, but most of the paper reads like you have continuous actions. This should be clarified early on that you assume discrete actions/ action-primitives for finding the morphologies.
Eq 1: j appears in the right but without any specification.
You write that you take expectations of joints, but what is p(j), so the probability of a joint. I think you want to add a sum over the joints $j$ or something. What is |J|?


Details:
- Sec 1: Second, Using (capital)
- Sec 3.1: Notice that left untouched, the...
- Sec 3.1: By assuming actions are uniform A_j?
- Action Distr: The appendix should contain information about the action sequences you are using.
- GNN Classifier: the the
- Sec 3.3: slowest step in for simulation....
- Sec 4.4: "meta" action: use the same label as in Fig 5  (global action)
- Table 3: ..TAME and a similar
- Sec 5: You write randomly sampled actions, but you have highly structured action-primitives

---

> ### Author Response · Authors · 2020-11-12
> **Response to Reviewer #1**
>
> We thank the reviewer for their insightful comments and for characterizing our work as “interesting” and “clear”. We have undergone another iteration of the paper to address the questions highlighted in the review and additionally added more ablations and baselines and are happy to continue iterating with your feedback.
>
> *> maybe more for a conference like ALife than ICLR though.*
>
> We believe there is ample precedent for work along this line at ICLR, given our supervised baseline, NGE (Wang et al 2019), was featured at ICLR 2019.
>
> *> the mathematical formulation is sloppy in many places*
>
> In the most recent version we have revised this formulation to make it more clear. We additionally encourage the reviewer to examine the revisions in the appendix where a revised full derivation can be found.
> 1. We add subscripts to the expectations to make it clear what the values and distributions the expectation is being taken over.
> 2. We clarify the text of the paper everywhere to make it clear that we deal with a discrete set of action primitives
> 3. *J appears in the right without specification… should be a sum…*: We reworked the derivation in the paper and left it in terms of expectation of overall actions. In the appendix we show how this can be decomposed into a sum and additionally reference this in section 3.2.
> 4. *What is |J|?*: -- this was a typo and was supposed to be log |A_j|. It has been corrected.
>
> We thank the reviewer for providing relevant references and ideas. We have included the citation to Klyubin, Polani, and Nehaniv in the related work.
>
> We have additionally made all corrections listed in the “details” portion of the review. Please let us know if you have any additional comments, questions, or concerns. Thank you!

---

> > ### Comment · AnonReviewer1 · 2020-11-21
> > **Update**
> >
> > Thank you for your answers and revisions. As my main concerns are addressed, I am upvoting my score.

---

### Official Review · AnonReviewer3 · 2020-10-29
**Official Blind Review #3**

**Rating:** 6
**Confidence:** 4

**Review:**

The paper presents an approach that evolves morphologies for virtual creatures that can reach as many states in their environment as possible. The approach is based on an information-theoretic objective that rewards morphologies for reaching diverse states and at the same time performing behaviours that are predictable.

While the paper is interesting there are a few issues that should be addressed:
- The approach relies on randomly sampled actions. How would it scale to more complex tasks that require non-random actions to be solved? Additionally, most of the morphologies are relatively simple, compared to some of the other work in evolving virtual creatures. Is it possible to scale the approach to produce more complex morphologies?
- "This is at odds with biological morphologies that seek to be functional over the largest subset of their environment as possible. " -> I would argue the opposite. Biological morphologies are very much specialist to work well in their particular niche.
- The approach does not seem completely task-agnostic since the actions are sampled from different distributions for each task (e.g. sine/cosine).
- The disadvantages and limitations of the approach should be discussed in more detail
- "We implement a variant of the task supervised Neural Graph Evolution (NGE) algorithm from Wang et al. (2019) without value based population pruning to better isolate the effects of reward supervision. “ -> how well does it work with value-based population training?
- How does the approach compare to quality diversity approaches that also aim to produce many morphologies in a single run, e.g. Novelty Search with Local Competition ("Evolving a Diversity of Virtual Creatures through Novelty Search and Local Competition" Lehman et al.)?

---

> ### Author Response · Authors · 2020-11-12
> **Response to Reviewer #3**
>
> We thank the reviewer for finding our approach interesting. We hope to provide clarifications and additional results to address the highlighted issues. Please let us know if there are further questions.
>
> *> The approach relies on randomly sampled actions. How would it scale to more complex tasks…*
>
> Our approach uses random actions to design morphologies, not necessarily complete the final task. We use random action primitives as a proxy for how modern RL algorithms explore, giving us an estimate of how good a given morphology will perform. If an agent is able to reach a large number of states with random actions, it's likely to be able to complete tasks involving that part of the state space. For example, the locomotion morphologies evolved by TAME would likely be able to complete goal reaching tasks. However, as you point out, this is not a guarantee. In scenarios where novel state space components are introduced far away from the initial state, TAME would be unlikely to discover them during its short primitive rollouts. We expand discussion at the end of section 4.4.
>
> *> The morphologies are relatively simple…*
>
> Our environments are similar to those of contemporary literature that use Mujoco (Wang et al) though with a few critical differences. While our environment  formulation is similar to that of NGE, unlike in NGE we consider contacts between joints. This assumption is more realistic, but makes it more difficult to add joints to locomotion agents. Despite this the number of joints in our evolved agents qualitatively appears similar. To our knowledge all publications besides NGE that use the mujoco simulator (Shaff et al 2019, Luck et al 2020, Chen et al 2020.) operate on fixed topologies. While we focused on algorithmic improvements, generative encodings for more complex morphologies is another field of research (Cheney, et al. 2013) independent of our contribution that has shown to lead to more complex structures.
>
> *> This is at odds with biological morphologies*
>
> Thanks for pointing this out. It is true that biological organisms evolve to fit their niche. Within this niche, however, we believe organisms develop multi-task behavior. TAME can be thought of fitting the morphology to the environment rather than the task. We have revised the paper to reflect this.
>
> *> The approach does not seem completely task-agnostic.*
>
> The action sequences do encode a bias towards the type of environment the agents are in, but not necessarily the task. However, the primitives are a) easy to select from the environment before task specification and b) aren’t the largest factor in performance. We have added a heading in section 4.4 discussing this. Empirically, we find that even using a different primitive space our method is still competitive to the NGE-like baselines in the locomotion environments.
>
> *> The disadvantages and limitations of the approach should be discussed in more detail*
>
> This has been done at the end of section 4.4
>
> *> how well does it [NGE] work with value-based population training?*
>
> We have run this baseline and added it to table 1. We find that value-based pruning helps performance a bit, though not extremely. Specification of the baseline is now given in section 4.2 and discussion in section 4.3.
>
> *> How does the approach compare to quality diversity approaches that also aim to produce many morphologies in a single run?*
>
> Our approach does not compete with quality diversity methods and is rather complementary to them. Quality diversity methods (Lehman & Stanley, 2011; Nordmoen et al., 2020) still require a task-based fitness function during evolution to select the best morphologies within “niches” or unsupervised clusters. The TAME objective could be substituted in for task-fitness in QD approaches to evolve a population of diverse agents in a completely unsupervised manner. We added citations for QD methods in the related work.

---

> > ### Comment · AnonReviewer3 · 2020-11-24
> > **Follow-up**
> >
> > Thank you for the detailed response to my questions. In general, I'm happy with the clarifications and increased my score (5->6). As some of the other reviewers also noted, the actions to explore the environment are rather simple and even though not crucial, there is some bias in the selection of primitives for the different tasks that work best. I'm certainly interested to see how this approach will work on more complex problems in the future.
> >
> > Minor comment:
> > “.... Moreover, all these methods are gradient-based, restricting them to fixed topology optimization where morphologies cannot have a varying number of joints.” -> This suggests that all the approaches in this paragraph are gradient-based, which is not the case. Maybe dividing it into two different paragraphs would help.

---

### Author Response · Authors · 2020-11-18
**Additional Evaluations Added**

Reviewers #2, #3, and #4 all raised questions regarding our methods ability to scale to more complicated tasks, with reviewer #2 specifically asking about a locomotion and reach task. Subsequently, we have added addition experimental evaluations of our method on a goal-reaching locomotion task in both 2D and 3D in appendix C of the paper.  We find that our method, without reward or task specification, is within confidence bounds of the NGE-like baselines in 2D and outperforms it in 3D. We hope this will quell concerns raised about the broader applicability of our approach.

For greater context, we believe it is also worth mentioning that most works in un-restricted topology morphology optimization do not consider tasks beyond forward locomotion. A lot of morphology literature uses forward locomotion as the primary task (Cheney et al 2013, Wang et al. 2019, Shaff et al 2019, Ha et al 2019, and Luck et al 2020). We not only consider the multi-task setting, but now consider both multi-directional locomotion and multi-directional locomotion and reach.

---

### Decision · Program_Chairs · 2021-01-07
**Final Decision**

**Decision:**

Accept (Poster)

**Comment:**

The authors use Empowerment for morphology optimisation, a quite novel idea. After initial unclarities and various improvements on the submission, the reviewers unanimously voted for acceptance of the paper.